

# HMGB1-activatied NLRP3 inflammasome induces thrombocytopenia in heatstroke rat

Huimei Yin[1], Ming Wu[2], Yong Lu[3,4], Xinghui Wu[4,5], BaoJun Yu[6], Ronglin Chen[7], JieFu Lu[8] and Huasheng Tong[4,5]

[1] The 3rd Xiangya Hospital, Central South University, Department of Critical Care Medicine and Hematology, Changsha, Hunan, China
[2] Department of Intensive Care Unit & Infection Prevention and Control, The Second People's Hospital of Shenzhen, Shenzhen, Guangdong, China
[3] Department of Critical Care Medicine, The First People's Hospital of Chenzhou, Chenzhou, Hunan, China
[4] Department of Graduate School, Southern Medical University, Guangzhou, Guangdong, China
[5] Department of Intensive Care Unit, General Hospital of Southern Theatre Command of PLA, Guangzhou, Guangdong, China
[6] Department of Intensive Care Unit, Baoan District People's Hospital, Shenzhen, Guangdong, China
[7] Department of Critical Care Medicine, Longgang District Central Hospital, Shenzhen, Guangdong, China
[8] Department of Intensive Care Unit, The First People's Hospital of Foshan, Foshan, Guangdong, China

Corresponding author
Huasheng Tong, fimmuths@163.com

## ABSTRACT

**Background**. Thrombocytopenia, an early common complication in heatstroke (HS), has been widely considered as a mortality predictor of HS. The mechanism underlying thrombocytopenia in HS remains unknown. It is not known whether NOD-like receptor family pyrin domain containing 3 (NLRP3) inflammasome is activated in HS platelet, which, in turn, induces platelet activation and thrombocytopenia. This study tried to clarify the activation of the NOD-like receptor signaling pathway under HS conditions and investigate its roles in mediating HS-induced thrombocytopenia.

**Methods**. Rat HS models were established in a certain ambient temperature and humidity. Platelets, isolated from blood, were counted and CD62P, an index of platelet activation, was measured by flow cytometry in all rats. The colocalization of NLRP3 inflammasome in platelet was detected by confocal fluorescence microscopy. Mitochondrial-derived reactive oxygen species (ROS) was detected using the molecular probes. Plasma HMGB1 and IL-1β levels were measured by ELISA.

**Results**. Platelet activation, showed by upregulated CD62P, and thrombocytopenia were observed in HS rats. HS activated the NLRP3 inflammasome, which was induced by elevated levels of ROS, while the upregulated CD62P and thrombocytopenia triggered by NLRP3 inflammasome were attributed to the high mobility group box protein 1 (HMGB1) inplasma. Moreover, inhibition of the NOD-like receptor signaling pathway in rats with HS suppressed platelet activation and the decline of platelet count. Similar results were obtained when the receptor toll-like receptor 4 (TLR4)/advanced glycation end product (RAGE) was blocked.

**Conclusions**. The NOD-like receptor signaling pathway induces platelet activation and thrombocytopenia in HS rats. These findings suggested that the NLRP3 inflammasome might be the potential target for HS treatment.

## BACKGROUND

Heatstroke (HS) is a life-threatening condition characterized by a core body temperature over 40.5 °C (*Epstein & Yanovich, 2019*). which was often complicated with multiple organ dysfunction syndrome (MODS) and even multiple organ failure (MOF), including acute hepatic failure (AHF), acute renal failure, disseminated intravascular coagulation (DIC), (*Hifumi et al., 2018*). Every few years, the deadly heat wave occurs and leads to thousands of deaths from HS (*Gaudio & Grissom, 2016*).

During the past few decades, evidences suggest that multi-organ damage correlates with the development of systemic inflammatory response syndrome (SIRS) that is stimulated by cytokines and other immune modulators in HS (*Epstein & Yanovich, 2019*). Platelets, mainly involved in hemostasis and thrombosis, also play important roles as inflammatory cells in innate and adaptive immune responses, which is possibly involved in the SIRS in HS (*Hifumi et al., 2018*; *Vieira-de Abreu et al., 2012*; *Semple, Italiano Jr & Freedman, 2011*). Importantly, thrombocytopenia, one of the most common complications frequently occuring early within 24 h after HS onset, has been taken as a good predictor of HS-related mortality (*Fan et al., 2015*). However, how about the pathophysiological responses of platelet and the pathogenesis of thrombocytopenia are to be elucidated in HS.

NOD-like receptor family pyrin domain containing 3 (NLRP3) inflammasome has been extensively investigated in the processes of dysbalanced inflammation and coagulation cascade (*Qiao et al., 2018*). NLRP3 inflammasome contributed to platelet activation and aggregation, and thrombosis via activating caspase-1 within inflammasome complexes (*Murthy et al., 2017*; *Qiao et al., 2018*). *Hottz et al. (2013)* found NLRP3 inflammasome was activated to release increased interleukin-1β (IL-1β) in activated platelets, with increased expression of P-selectin (CD62P), which is associated with thrombocytopenia in patients with dengue fever. IL-1β, an important inflammatory factor mediated by NLRP3-dependent caspase-1 activation, is linked to increased endothelial permeability (*Bozza et al., 2008*), thrombosis, and dysregulated hemostasis in dengue fever (*Qiao et al., 2018*). What's more, significantly increased IL-1β levels have been found in HS (*Geng et al., 2015*).

High mobility group box protein 1 (HMGB1), a typical damage-related molecular pattern molecule, is often transferred from the nucleus to the cytoplasm and can be extracellularly released under many stimuli (*Kang et al., 2014*). We have previously demonstrated that early elevated plasma levels of HMGB1 prognosed the severity and mortality in patients with HS (*Tong et al., 2011*). More importantly, *Geng et al. (2015)* found that HS induced liver dysfunction via HMGB1-induced activation of NLRP3 inflammasome and IL-1β release. It is well known that HMGB1 plays roles in regulating platelet activation, microparticle secretion and adhesion, and thrombus formation (*Vogel et al., 2015*; *Vallance et al., 2017*). Moreover, mice lacking HMGB1 in platelets exhibited

increased bleeding time, reduced thrombus formation, platelet aggregation, inflammation, and organ damage during experimental trauma/hemorrhagic shock (*Vogel et al., 2015*). Unfortunately, whether HMGB1-induced activation of NLRP3 inflammasome is involved in platelet activation and thrombocytopenia in HS is to be elucidated.

In this study, using a rat model of HS, we tried to investigate whether HMGB1-induced activation of NLRP3 inflammasome occurs via its receptors including toll-like receptor 4 (TLR4) and the receptor for advanced glycation end product (RAGE) signaling pathway, and contributes to platelet activation and thrombocytopenia.

## MATERIALS & METHODS

### Experimental animals

Because estrogen has certain effects on HS pathogenesis and organ injury (*Chen et al., 2006*), only adult male, pathogen-free Sprague-Dawley (SD) rats (Experimental Animal Center of the General Hospital of Southern Theatre Command, license number for animal experimentation: SCXK, Guangdong 2016-0041), weighing 220 to 250 g, were used in this study. The rats received humane care according to the criteria outlined in the Guide for the Care and Use of Laboratory Animals (National Institutes of Health publication 86-23, 1985 revision). We adhered to the guidelines for animal care of the General Hospital of Southern Theatre Command. The animal ethics committee of the General Hospital of Southern Theatre Command of PLA approved this study.

### Rat model of HS

Rats housed for 6 h at ambient temperature ($25\,°C \pm 0.5\,°C$) with a humidity of $35\% \pm 5\%$, 126 rats were randomly divided into 21 groups, with 6 rats per group. Animals have free access to standard food and water. To induce HS, rats were placed in a prewarmed incubator maintained at $39.5\,°C \pm 0.2\,°C$ with a relative humidity of $60.0\% \pm 5.0\%$. The rectal temperature (Tr) was monitored at 10-min intervals using a thermocouple (BW-TH1101; Biowill, Shanghai, China), which was inserted 5.5 cm into the rectum of each rat. The time point at which the Tr reached $43\,°C$ was considered a reference point of HS onset (*Geng et al., 2015*). The rats were removed from the incubator after HS onset, transferred to room temperature ($22.0\,°C \pm 0.5\,°C$), and fed with adequate food and water. The rats in control group were sham-heated at a temperature of $25\,°C \pm 0.5\,°C$ and a humidity level of $35\% \pm 5\%$. Ethyl pyruvate (EP, 50 mg/kg body weight, MCE, USA) was injected intraperitoneally (i.p.) to inhibit HMGB1 release, or the same volume phosphate buffered solution (PBS) were injected as a negative control before rats subjected to heat stress. A neutralizing antibody specific for rats against HMGB1 (3 mg/kg body weight, Santa Cruz Biotechnology, Santa Cruz, CA, USA) was immediately injected i.p. into the rats before heat stress. In some rats, TLR4 neutralizing antibody specific for rats (8 μg/kg body weight, Abcam, Cambridge, MA, USA), RAGE neutralizing antibody (80 μg/kg body weight, R&D, USA), or both were injected i.p. before rats subjected to heat stress. Other rats were injected i.p. with Ac-Tyr-Val-Ala-Asp-chloromethylketone (ac-YVAD-cmk, 0.3 mg/kg body weight, Enzo Biochem Inc, New York, USA) which is the caspase-1 inhibitor or PBS containing 2.8% dimethyl sulfoxide (DMSO) as a control before rats subjected to

heat stress. Some rats were given MCC950 (20 mg/kg body weight, Selleckchem, Houston, TX, USA), a specific blocker of NLRP3, 1 h before heat stress. Some rats were injected i.p. with the antioxidant N-acetylcysteine (NAC, 300 mg/kg body weight, MCE) immediately before heat stress. At the end of the experiment, rats were anesthetized by inhalation of isoflurane, and sacrificed to collect blood from abdominal aorta.

## Platelet count
50 μL of whole blood was collected into a sodium citrate anticoagulation tube to measure the platelet count using a blood cell analyzer (BC-3000; Mairui, Shenzhen, China).

## Platelet isolation
Platelets were isolated as described previously (*Qiao et al., 2018*). Briefly, three mL of anticoagulant blood samples were centrifuged at 150 g for 15 min to obtain platelet-rich plasma (PRP), and about 3/4 of the PRP was carefully pipetted into two volumes of modified Tyrode buffer (12 mM $NaHCO_3$, 138 mM NaCl, 5.5 mM glucose, 2.9 mM KCl, 2 mM $MgCl_2$, 0.42 mM $NaH_2PO_4$, 10 mM HEPES, pH 7.4). The mixture was centrifuged at 37 °C at 500 g for 8 min in the presence of PGE1 (0.1 μg/mL; Cayman Chemical) and apyrase (1 U/mL, Sigma-Aldrich), washed twice with CGS buffer (120 mM sodium chloride, 12.9 mM trisodium citrate, 30 mM D-glucose, pH 6.5), and resuspended in modified Tyrode buffer.

## Transmission electron microscopy examination
One volume of PRP prepared above was added into 45 volumes of 3% glutaraldehyde and to stand for 4 h. The samples were centrifuged at 800 g for 10 min and the supernatant was discarded. The pellet was washed with 0.1 M phosphate buffer for twice, postfixed with 1% osmium tetroxide at 4 °C for 1 h, and dehydrated in a graded ethanol series and embedded in Epon. Epon-embedded platelets were cut into sections (70–80 nm) with an Ultracut UCT ultramicrotome, transferred to copper grids, and stained with uranyl acetate and lead citrate. The sections were then observed with transmission electron microscope (TEM) (H-7560, Hitachi, Japan).

## Flow cytometric analyses
Freshly isolated platelets ($10^6$ to $10^7$) were resuspended in one mL of modified Tyrode buffer, and treated with 0.2% triton before to measure intracellular cytokines. A minimum of 10,000 events per gate was acquired using a flow cytometer (BD LSR Fortessa). P-selectin (CD62P) was determined using the anti-CD62P monoclonal antibody (1:1000; Biolegend, USA), and TLR2 (toll-like receptor 2), TLR4, and RAGE were assessed with antibodies targeted to TLR2 (1:500; Abcam), TLR4 (1:1000; Abcam), and RAGE (1:1000; Abcam). NLRP3 expression was evaluated by an NLRP3 antibody (1:1000; Abcam), and activation of caspase-1 was assessed by the fluorescent probe green fluorescent-labeled inhibitor of caspase-1 (FLICA, 1:100, Immunochemistry Technologies, Bloomington, MN, USA), which irreversibly binds to activated caspase-1. IL-1β was evaluated by the anti–IL-1β antibody (1:1000; Biorbyt). Mitochondrial-derived reactive oxygen species (ROS) were detected using the molecular probes Mito-SOX (for mitochondria $O_2\bullet$-, 1:1000; Invitrogen), DHE (for cytoplasmic $O_2\bullet$-, 1:1000; Invitrogen), and DCF-DA (for

H$_2$O$_2$, 1:1000; Invitrogen). All indices were doubly labeled with a phycoerythrin- or fluorescein isothiocyanate–conjugated CD61 monoclonal antibody (1:1000; Biolegend) to label platelets. The results were analyzed using FlowJo software 7.6.

## Confocal fluorescence microscopy

Immunofluorescence for NLRP3 and apoptosis-associated speck-like protein containing a caspase-recruitment domain (ASC) was performed according to previous research (*Feng et al., 2014*). Platelets attached on a poly-L-lysine-coated coverslip were fixed with pre-cooled methanol for 20 min, washed twice with PBS (135 mM NaCl, 4.7 mM KCl, 10 mM Na$_2$HPO$_4$, 2 mM NaH$_2$PO$_4$, pH 7.4), and blocked with 10% fetal calf serum (FCS) in PBS. Then, platelets were incubated with goat anti-rat NLRP3 antibody (1:1000; Abcam) or rabbit anti-rat ASC antibody (1:500; Santa Cruz) for 30 min, washed twice with PBS, and labeled with corresponding Dlight 488 and Dlight 594 conjugated secondary antibodies (1:1000; Invitrogen) for 30 min respectively. Preparations were analyzed under a laser scanning confocal microscope (Fluo View FV1000; Olympus), and FV1000 operations software was used for recording.

## Plasma HMGB1 and IL-1β levels measurement

According to the manufacturer's instructions, plasma HMGB1 levels were measured by a commercially available HMGB1 enzyme-linked immunosorbent assay kit (Shino-Test Corporation, Sagamihara, Japan), and IL-1β levels were measured by a commercially available IL-1β enzyme-linked immunosorbent assay kit (Invitrogen, Waltham, MA, USA).

## Statistical analysis

All data were presented as the mean $\pm$ SD unless stated otherwise. All data followed a normal distribution except for one group, which was analyzed with Kruskall-Wallis. For the data followed a normal distribution, statistical significance was determined with the least significant difference $t$-test or one-way analysis of variance (ANOVA). IBM SPSS Statistics 26 software was used for data. $P < 0.05$ was considered statistically significant.

# RESULTS

## HS induces platelet activation and thrombocytopenia in HS rats

The platelet count began to decline at 3 h after HS onset (Fig. 1A), whereas the expressions of IL-1β (Fig. 1D) and CD62P (Fig. 1E) in platelets increased progressively during the same period after HS onset. All changes were peaked at 9 h after HS onset. The platelet parameters such as mean platelet volume (MPV) and platelet distribution width (PDW) were assessed. MPV (Fig. 1B) and PDW (Fig. 1C) at 3 h and 6 h after HS maintained in normal ranges, with no statistical differences comparing to the sham group. However, both MPV and PDW at 9 h after HS increased significantly compared to those in the control group after 9 h of HS ($P < 0.01$). Therefore, for the remaining experiments, we observed heat-stressed animals only at the time point of 9 h after HS onset.

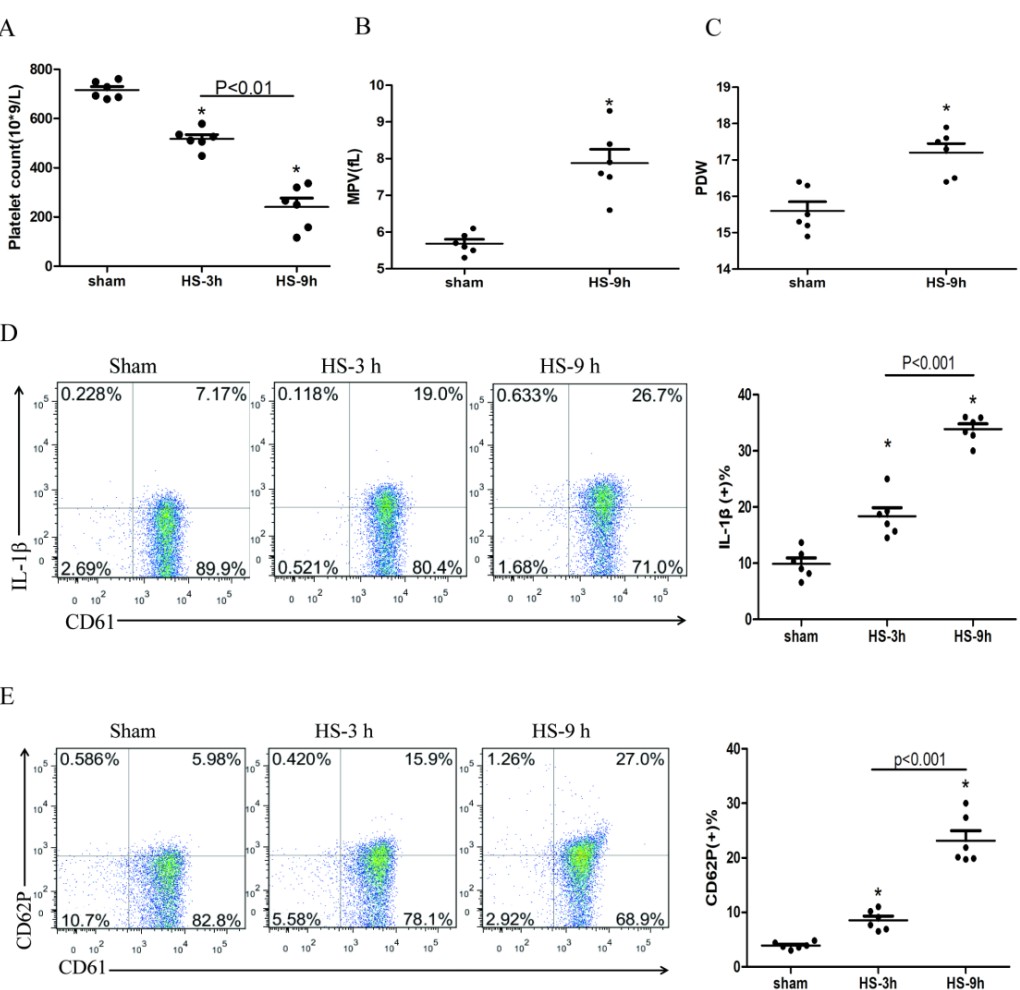

**Figure 1  HS induces platelet activation and thrombocytopenia in HS rats ($n = 6$).** (A) Thrombocytopenia in HS rats. *$p < 0.001$ *vs* sham. (B) Changes of MPV in HS rats. *$p < 0.001$ *vs* sham. (C) Changes of PDW in HS rats. *$p = 0.001$ *vs* sham. (D) Increased expression of IL-1 c in HS rats. *$p < 0.001$ *vs* sham. (E) Increased expression of CD62P in HS rats. $p = 0.003$ between HS-3 h *vs* sham, $p < 0.001$ between HS-9 h *vs* sham, $p = 0.01$ between HS-3 h *vs* HS-6 h.

## NLRP3 inflammasome activation in platelets in HS rats

The association of NLRP3 and ASC as well as caspase-1 represents activation of the inflammasome. The colocalization of NLRP3 (green) and ASC (red) in platelets was observed under a confocal microscopy (Fig. 2C). In addition, flow cytometric analyses showed that NLRP3 (Fig. 2A) and cleaved caspase-1 (Fig. 2B) increased at 3 h and reached the peak levels at 9 h after HS onset. All results indicate the HS-inducing NLRP3 inflammasome activation in platelets.

## NLRP3 inflammasome mediates platelet activation and thrombocytopenia in HS rats

To investigate whether HS-activated NLRP3 inflammasome mediates platelet activation and thrombocytopenia, we injected rats i.p. with MCC950, a specific inhibitor of the

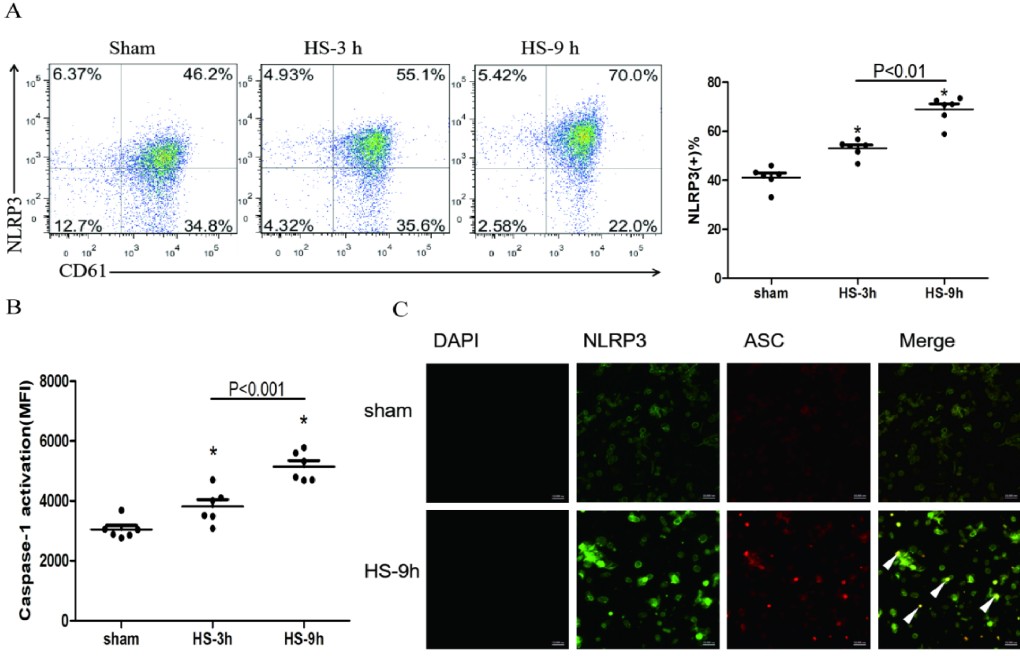

**Figure 2** **NLRP3 inflammasome activation in platelets in HS rats (*n* = 6).** (A) NLRP3 expression in HS rats. *$p < 0.001$ *vs* sham. (B) caspase-1 activation in HS rats. *$p < 0.001$ *vs* sham. (C) NLRP3 Inflammasome assembly in platelets of rat at 9 h after HS onset. NLRP3 (green) and ASC (red) and Merge (yellow) were observed under a laser confocal microscope. The white triangular arrows point to the NLRP3 inflammasome, suggesting the presence of NLRP3 and ASC in platelets from rats with HS. Bars represent 10,000 nm.

NLRP3 inflammasome, before the rats subjected to heat stress. The results showed that inhibited NLRP3 inflammasome significantly down-regulated the levels of cleaved caspase-1 (Fig. 3A), IL-1β secretion (Fig. 3B) and CD62P expression (Fig. 3C) and reduced the drop of platelet count (Fig. 3D) at 9 h after HS onset. Moreover, pharmacological inhibition of caspase-1 activity with ac-YVAD-cmk also markedly down-regulated the IL-1β secretion (Fig. 3E) and CD62P expression (Fig. 3F) as well as the drop of platelet count (Fig. 3G) at 9 h after HS onset.

## HMGB1 activates platelet NLRP3 inflammasome in HS rats

Previous studies found that early increased extracellular HMGB1 levels in HS were negatively correlated with its prognosis (*Tong et al., 2013a*). HMGB1 activates the NLRP3 inflammasome in liver cells, which in turn promotes hepatocyte pyroptosis (*Geng et al., 2015*). Consistent with previous studies (*Tong et al., 2011*), we found that plasma HMGB1 concentrations in HS rats were gradually increased (Fig. 4A). We also unraveled the role of HMGB1 in inducing NLRP3 inflammasome activation, platelet activation, and thrombocytopenia in HS. We injected rats i.p. with EP to inhibit HMGB1 release and used anti-HMGB1 neutralizing antibody to block the effect of HMGB1 before submitting rats to HS.

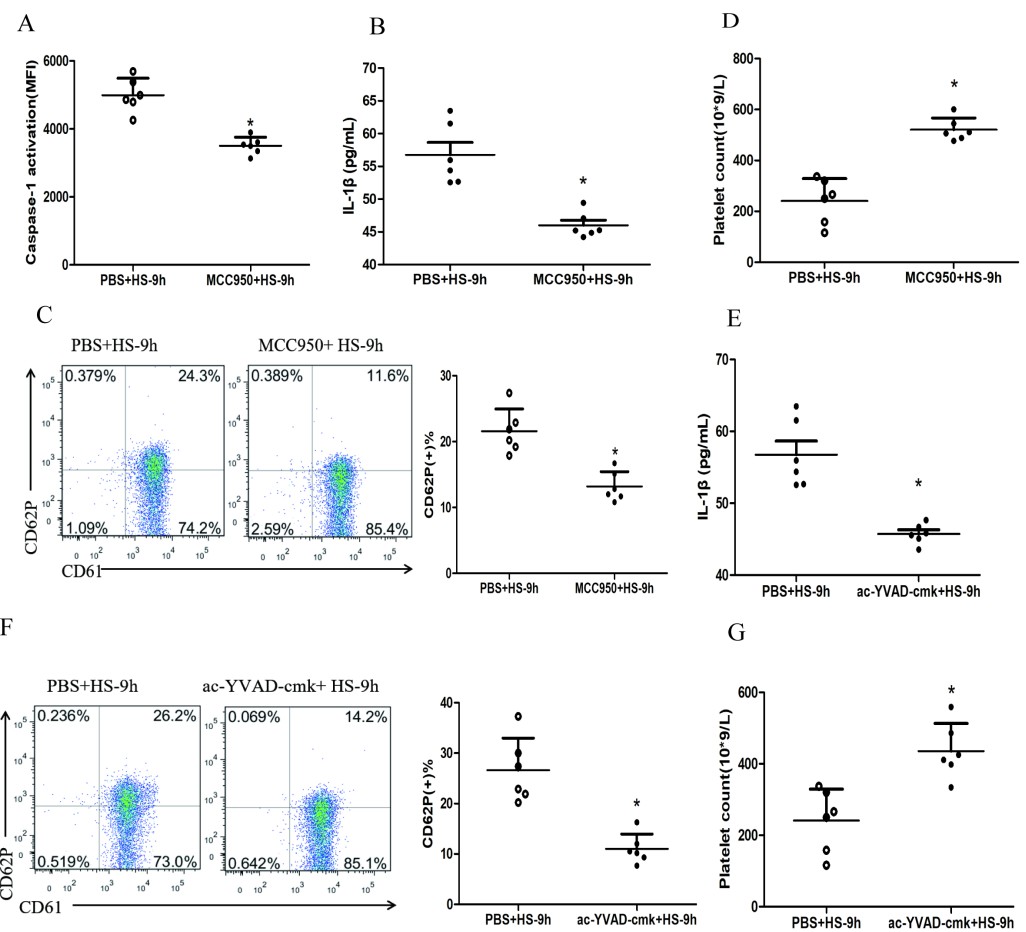

**Figure 3** **NLRP3 inflammasome mediates platelet activation and thrombocytopenia in HS rats ($n = 6$).** (A) Effect of MCC950 on caspase-1 activation in platelet of HS-9 h rats. *$P < 0.001$ vs PBS+HS-9 h. (B) Effect of MCC950 on IL-1β secretion in HS-9 h rats. *$P = 0.001$ vs PBS+HS-9 h. (C) Effect of MCC950 on CD62P in platelet of HS-9 h rats. * $P < 0.001$ vs PBS+HS-9 h. (D) Effect of MCC950 on platelet count of HS-9 h rats. *$P < 0.001$ vs PBS+HS-9 h. (E) Effect of ac-YVAD-cmk on IL-1β secretion in HS-9 h rats. *$P < 0.001$ vs PBS+HS-9 h. (F) Effect of ac-YVAD-cmk on CD62P in platelet of HS-9 h rats. *$P < 0.001$ vs PBS+HS-9 h. (G) Effect of ac-YVAD-cmk on platelet count of HS-9 h rats. *$P = 0.002$ vs PBS+HS-9 h.

EP inhibiting or neutralizing antibody against HMGB1 significantly reduced the expression of NLRP3 (Fig. 4B) and cleaved caspase-1 (Fig. 4C) at 9 h after HS onset. Notably, levels of secreted IL-1β (Fig. 4D), expression of CD62P (Fig. 4E), and thrombocytopenia (Fig. 4F) at 9 h after HS onset were significantly alleviated after pretreatment with EP and anti-HMGB1 antibody. These results indicate that HMGB1 plays an important role in activating NLRP3 inflammasome and possibly participates in platelet activation and thrombocytopenia in HS.

In the early stages of sterile inflammation, HMGB1 activates the NLRP3 inflammasome through TLR4 or RAGE receptors (*Xu et al., 2013*). As shown in Figs. 4G–4I, expressions of TLR2, TLR4, and RAGE in platelets were gradually increased after HS onset. To define the receptors that mediate HMGB1-induced NLRP3 inflammasome activation, we applied

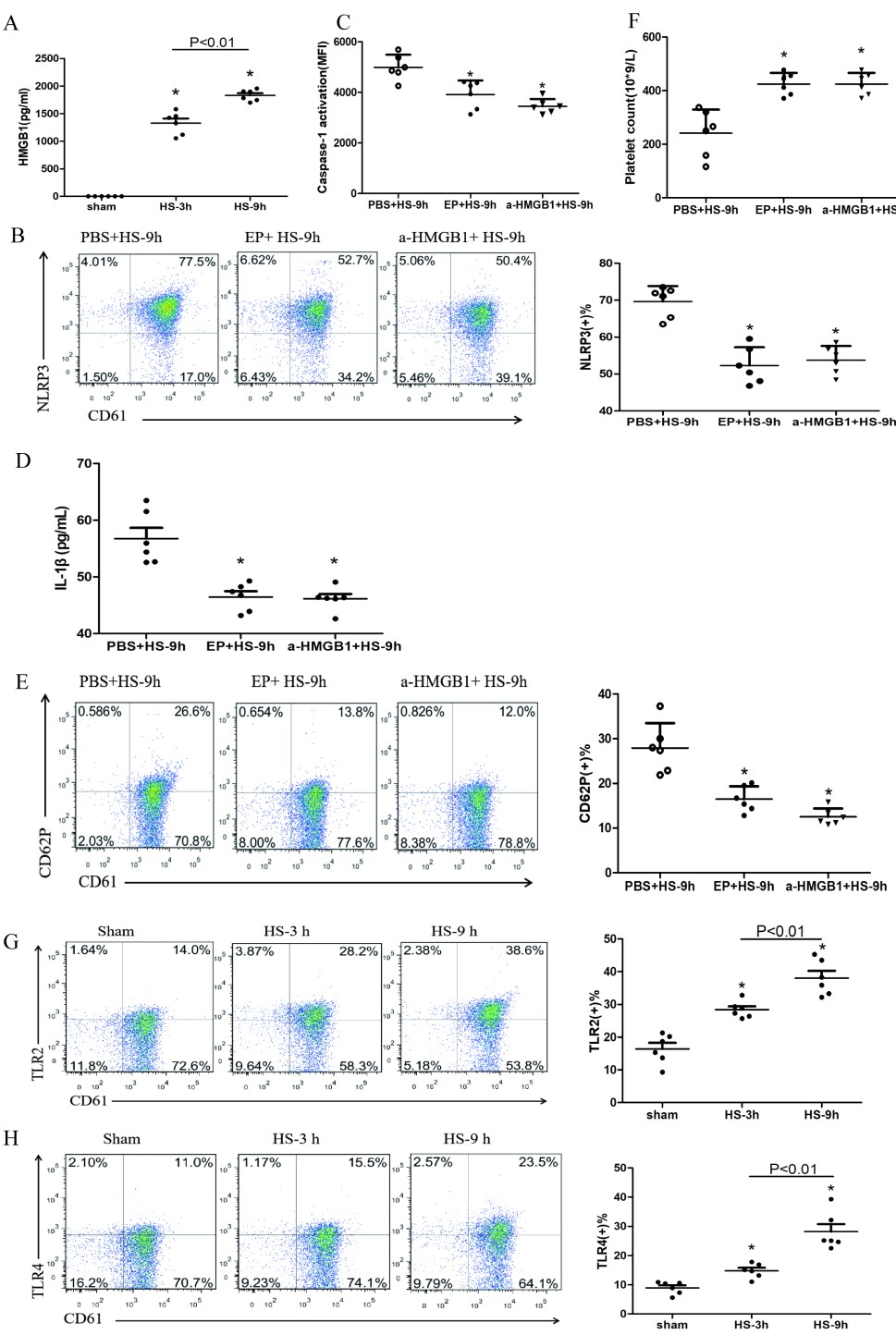

**Figure 4  HMGB1 activates platelet NLRP3 inflammasome in HS rats ($n = 6$).** (A) HMGB1 levels in plasma of HS rats. *$P < 0.001$ *vs* sham, $P = 0.003$ between HS-3 h *vs* HS-6 h. (B) Effects of EP and a-HMGB1 on NLRP3 expression in platelets of HS-9 h rats. *$P < 0.001$ *vs* PBS+HS-9 h. 

**Figure 4 (…continued)**
(C) Effects of EP and a-HMGB1 on caspase-1 activation in platelets of HS-9 h rats. *$P < 0.001$ *vs*
PBS+HS-9 h. (D) Effects of EP and a-HMGB1 on IL-1β secretion in HS-9 h rats. *$P = 0.001$ *vs* PBS+HS-9
h. (E) Effects of EP and a-HMGB1 on CD62P expression in platelets of HS-9 h rats. *$P < 0.001$ *vs*
PBS+HS-9 h. (F) Effects of EP and a-HMGB1 on platelet counts in HS-9 h rats. *$P < 0.001$ *vs* PBS+HS-9
h. (G) TLR2 expression in platelets of HS rats. *$P < 0.001$ *vs* sham. (H) TLR4 expression in platelets
of HS rats. $P = 0.004$ between HS-3 h *vs* sham, $p = 0.001$ between HS-9 h *vs* sham, $p = 0.007$ between
HS-3 h *vs* HS-6 h. (I) RAGE expression in platelets of HS rats. *$P = 0.001$ *vs* sham. (J) Effects of a-TLR4
and a-RAGE on NLRP3 expression in platelets of HS-9 h rats. * $P < 0.001$ *vs* PBS+HS-9 h. (K) Effects
of a-TLR4, a-RAGE on caspase-1 activation in platelets of HS-9 h rats. *$P < 0.001$ *vs* PBS+HS-9 h. (L)
Effects of a-TLR4, a-RAGE on IL-1β in platelets of HS-9 h rats. *$P < 0.001$ *vs* PBS+HS-9 h. (M) Effects of
a-TLR4, a-RAGE on CD62P in platelets of HS-9 h rats. $P = 0.003$ between a-TLR4+HS-9 h *vs* PBS+HS-9
h, $P = 0.003$ between a-RAGE+HS-9 h *vs* PBS+HS-9 h, $p < 0.001$ between a-TLR4+a-RAGE+HS-9 h
*vs* PBS+HS-9 h, $p = 0.269$ between a-TLR4+a-RAGE+HS-9 h *vs* a-TLR4+HS-9 h, $p = 0.242$ between
a-TLR4+a-RAGE+HS-9 h *vs* a-RAGE+HS-9 h. (N) Effects of a-TLR4, a-RAGE on platelet count in HS-9 h
rats. *$P < 0.001$ *vs* PBS+HS-9 h.

anti-TLR4 and anti-RAGE neutralizing antibodies before heat exposure. Either anti-TLR4
or anti-RAGE neutralizing antibody significantly inhibited HS-induced NLRP3 (Fig. 4J)
and cleaved caspase-1 (Fig. 4K) expression in platelets at 9 h after HS onset. In addition, the
combination of anti-TLR4 and anti-RAGE neutralizing antibody induced greater inhibition
of NLRP3 and cleavage of caspase-1 expression. Synchronous changes were also observed
in HMGB1-induced upregulated expression of IL-1β (Fig. 4L), and thrombocytopenia (Fig.
4N) at 9 h after HS. Anti-TLR4 or anti-RAGE neutralizing antibody inhibited HS-induced
CD62P expression, while combination of anti-TLR4 and anti-RAGE neutralizing antibody
made no significantly effect compared with single neutralizing antibody (Fig. 4M). These
data indicate that HMGB1 activates NLRP3 inflammasome via both TLR4 and RAGE
receptors, which is possibly associated with platelet activation and thrombocytopenia in
HS.

## HMGB1 activates platelet NLRP3 inflammasome by upregulating ROS in HS rats

ROS mainly includes the superoxide anion ($O_2-$), hydrogen peroxide ($H_2O_2$), and the
hydroxyl radical (HO–) (*Leytin, 2012*). Many studies confirmed HMGB1 could up-regulate
ROS (*Tsung et al., 2007*; *Pietraforte et al., 2014*), which was the key to activate NLRP3
inflammasome (*Dostert et al., 2008*; *Zhou et al., 2011*). To study the changes of ROS in
platelets and the effects of ROS on the HMGB1-induced NLRP3 inflammasome in HS,
we used a fluorescence probe to detect the expression of mitochondrial $O_2\bullet$-, cytoplasmic
$O_2\bullet$-, and $H_2O_2$ in platelets in HS rats. We found that mitochondrial $O_2\bullet$- (Fig. 5A),
cytoplasmic $O_2\bullet$- (Fig. 5B), and $H_2O_2$ (Fig. 5C) increased at 3 h after HS onset. Inhibiting
HMGB1 significantly reduced the level of ROS, which indicated that HMGB1 was involved
in high-level of ROS in HS (Figs. 5D–5F).

To explore the relationship between ROS and the NLRP3 inflammasome activation,
we pretreated rats with the antioxidant NAC. In platelets, NAC significantly inhibited
the expression of NLRP3 (Fig. 5G) and cleaved caspase-1 (Fig. 5H) at 9 h after HS onset.
Similar results were observed for secreted IL-1 β (Fig. 5I), CD62P expression (Fig. 5J), and

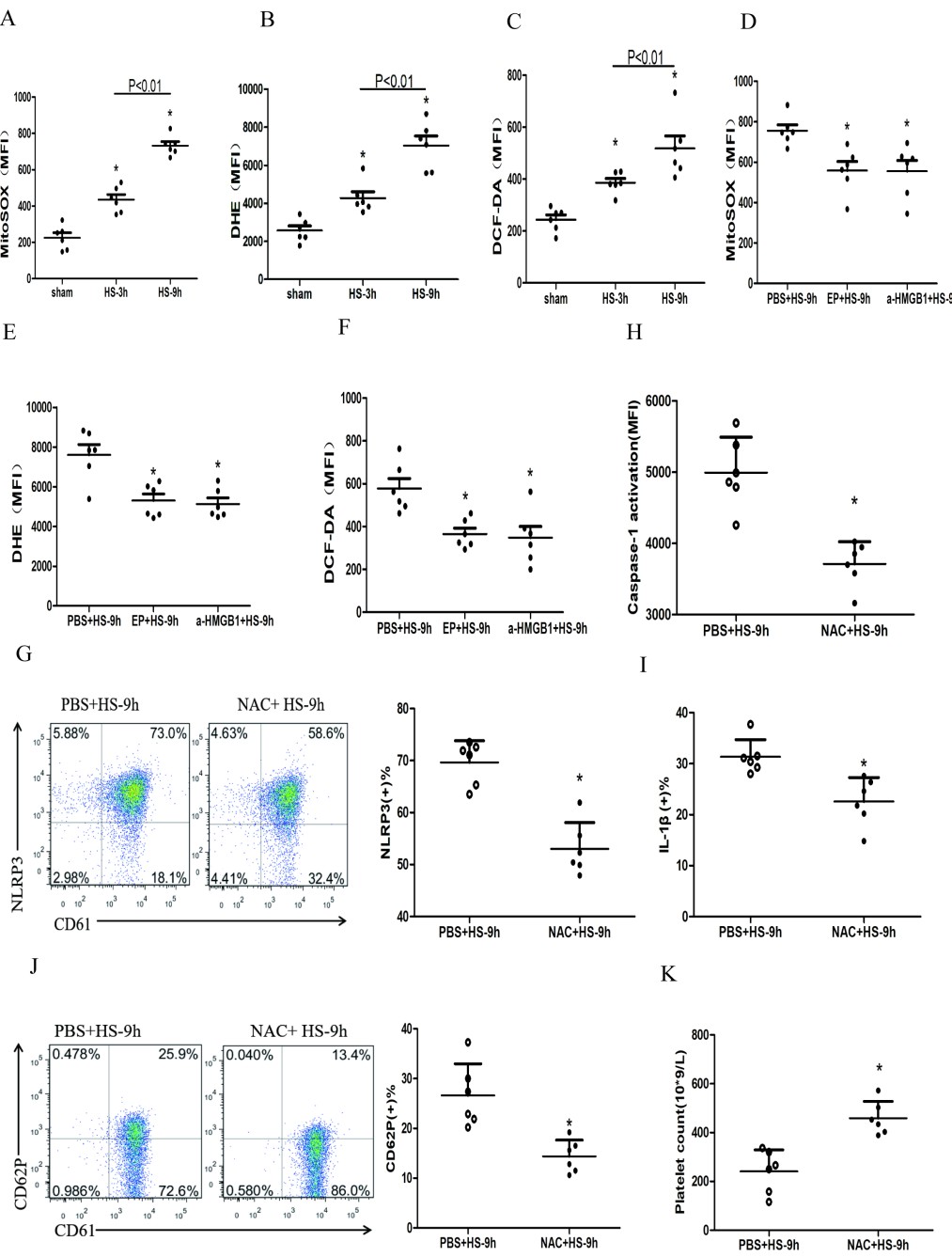

**Figure 5 HMGB1 activates platelet NLRP3 inflammasome by upregulating ROS in HS rats ($n = 6$).** (A) Mito-SOX levels in HS rats. $^*p < 0.001$ *vs* sham. (B) DHE levels in HS rats. $^*p < 0.001$ *vs* sham. (C) DCF-DA levels in HS rats. $^*p < 0.001$ *vs* sham. (D) Effects of EP and a-HMGB1 on Mito-SOX in platelets of HS-9 h rats. $^*P = 0.007$ *vs* PBS+HS-9 h. (E) Effects of EP and a-HMGB1 on DHE in platelets of HS-9 h rats. $^*P = 0.001$ *vs* PBS+HS-9 h. (F) Effects of EP and a-HMGB1 on DCF-DA in platelets of HS-9 h rats. $^*P = 0.003$ *vs* PBS+HS-9 h. (G) Effects of NAC on NLRP3 in platelets of HS-9 h rats. $^*P < 0.001$ *vs* PBS+HS-9 h. (H) Effects of NAC on caspase-1 activation in platelets of HS-9 h rats. $^*P < 0.001$ *vs* PBS+HS-9 h. (I) Effects of NAC on IL-1β secretion in HS-9 h rats. $^* P = 0.005$ *vs* PBS+HS-9 h. (J) Effects of NAC on CD62P in platelets of HS-9 h rats. $^*P = 0.002$ *vs* PBS+HS-9 h. (K) Effects of NAC on platelet count of HS-9 h rats. $^*P = 0.001$ *vs* PBS+HS-9 h.

thrombocytopenia (Fig. 5K). These results indicate that ROS is responsible for platelet NLRP3 inflammasome-induced activation and thrombocytopenia in HS.

## DISCUSSION

In the current study, we investigated the NOD-like receptor signaling pathway of platelet in HS rats, and explored the mechanisms involved in platelet activation and thrombocytopenia. The main results of our investigation were as following: (1) HS activatied the NOD-like receptor signaling pathway, which induced thrombocytopenia and platelet activation in HS rats. (2) High levels of extracellular HMGB1, through TLR4 and RAGE pathway, induced NLRP3 inflammasome activation via ROS possibly in HS rats.

MPV and PDW are indices of average platelet volume and variation in platelet size respectively, both of which are markers of platelet activation and thrombocytopenia (*Pogorzelska et al., 2020*). In this experiment, we found that MPV and PDW did not change at 3 h but were increased significantly at 9 h after HS onset. Previous study showed no evidence of megakaryocyte damage (*Gader, 1992*), We found platelet aggregation gradually decreased after HS onset (Fig. SA), which consists with the results that hyperthermia inhibit platelet aggregation (*Etulain et al., 2011*), indicating that peripheral platelet destruction may be a possible explanation of the thrombocytopenia of HS.

*Zhuang et al. (2011)* found that heat stress facilitated the procession of Px-caspase-1 (a caspase gene) expression, an important effector molecule of the NLRP3 inflammasome. *Murthy et al. (2017)* found that NLRP3 inflammasome affects platelet activation, aggregation, and thrombosis. Additionally, one team in our institute previously found that HMGB1-activatied NLRP3 inflammasome in liver cells during HS was associated with liver damage (*Geng et al., 2015*). Therefore, it was speculated that the HMGB1-activatied NLRP3 inflammasome possibly mediates platelet activation and thrombocytopenia in HS. We observed that the expressions of NLRP3, ASC, and cleaved caspase-1 in platelets of HS rats were increased at early stage after HS onset and remained increased thereafter (Figs. 2A, 2B and 2C). All these results reflect the assembly and activation of NLRP3 inflammasome in HS. Consistent with the results of a previous study (*Murthy et al., 2017*), we found that inhibiti NLRP3 and cleavage of caspase-1 alleviated platelet activation and thrombocytopenia (Fig. 3), indicating that the NLRP3 inflammasome played an important role in platelet activation and thrombocytopenia possibly.

HMGB1, a class of non-histone chromosomal binding protein presenting in eukaryotic cells, has been extensively studied as an important molecule activating NLRP3 inflammasome(*Park et al., 2006*). Our previous studies showed that plasma HMGB1, significantly elevated at an early stage in HS patients, was associated with more critical disease and poor prognosis (*Tong et al., 2011*). Inhibiting HMGB1 release or blocking HMGB1 alleviated liver sinusoidal endothelial damage (*Tong et al., 2013a*) and reduced thrombosis in HS (*Murthy et al., 2017*). In this experiment, we also found that inhibiting HMGB1 by EP and anti-HMGB1 neutralizing antibody down-regulated the activation of the NLRP3 inflammasome, platelet activation, and thrombocytopenia. Previous studies

have showed that HMGB1 promotes inflammation via the main receptors including TLR2,TLR4 and RAGE receptors (*Park et al., 2006*), TLR4-dependent upregulation of the platelet NLRP3 inflammasome promotes platelet aggregation (*Vogel et al., 2019*). Because there was no anti-TLR2 neutralizing antibody for rats, we only tested the effects of TLR4 and RAGE receptors. Different to the results of a prior study that visfatin-induced disruption on junction proteins of mouse vascular endothelial cells was conducted through the HMGB1-RAGE rather than the HMGB1-TLR4 pathway (*Chen et al., 2015*), our results indicate that both TLR4 and RAGE receptor-specific antibodies can inhibit NLRP3 inflammasome activation, platelet activation and thrombocytopenia. Also, anti-TLR4 antibody pretreatment significantly shortened the heat exposure time of rats (Fig. SB). These results suggest that HMGB1 activates the NLRP3 inflammasome via TLR4 and RAGE receptors and participates in platelet activation and thrombocytopenia possibly.

ROS, taking an important role in the homeostasis of aging and death of platelets (*Pietraforte et al., 2014*), are involved in platelet activation, adhesion, and aggregation and apoptosis (*Leytin, 2012*). Consistent with previous studies, three kinds of ROS in the platelets of rats with HS increased progressively with time (Figs. 5A, 5B and 5C) in our experiments. HMGB1 can upregulate intracellular levels of ROS in many diseases including HS (*Tang et al., 2011*). Similarly, we found inhibition or blockade of HMGB1 could partially inhibit the production of all the three kinds of ROS (Figs. 5D and 5E) in HS rats. It was found that HMGB1-induced elevated intracellular ROS (*Abais et al., 2015*; *Han et al., 2015*)  could promote NLRP3 inflammasome activation. We found that NAC pretreatment could inhibit the expression of platelet NLRP3 and cleaved caspase-1 as well as platelet activation and thrombocytopenia in HS rats, indicating that ROS could activate NLRP3 inflammasome in platelets and may be invovled in thrombocytopenia.

Pyroptosis is a caspase-1 dependent programmed cell death (*Geng et al., 2015*). In the current study, thrombocytopenia and active caspase-1 showed opposite trends, suggesting that there must be a link between them. Recent studies have shown that pyroptosis is a major cell death pathway of platelets treated with dengue virus and envelope protein domain III (*Lien et al., 2021*). Using different methods to validate the pyroptotic results, they they examined cleaved caspase-1, cleaved GSDMD and cell–surface GSDMD levels, IL-1β and lactic dehydrogenase release (*Lien et al., 2021*). However, we did not conduct experiments to verify pyroptosis in platelet of HS. whether pyroptosis is a kind of cell death pathway of platelets in the HS rat model need further research. In addition, we did not exclude the effect of direct thermal injury to the platelet. Furthermore, other signaling pathways in platelet may also be activated and involved in the inflammation process. Finally, our results did not fully reflect the effect of NLRP3 inflammasome on HS platelets for using rat models not patients.

## CONCLUSIONS

In conclusion, this study preliminarily demonstrates that elevated extracelluar HMGB1 induced high levels of ROS via both TLR4 and RAGE receptors of platelets, which in turn participates NLRP3 inflammasome activation leading to thrombocytopenia in HS rats. The recognition of this study may aid to develop novel therapeutic agents in future.

### Funding

This work was supported by the the PLA Logistics Research Project of China (18CXZ032) and the Natural Science Foundation of Guangdong Province (Grant No. 2022A1515010353). The funders had no role in study design, data collection and analysis, decision to publish, or preparation of the manuscript.

### Competing Interests

The authors declare there are no competing interests.

### Author Contributions

- Huimei Yin conceived and designed the experiments, performed the experiments, analyzed the data, prepared figures and/or tables, authored or reviewed drafts of the article, and approved the final draft.
- Ming Wu analyzed the data, prepared figures and/or tables, authored or reviewed drafts of the article, and approved the final draft.
- Yong Lu performed the experiments, analyzed the data, prepared figures and/or tables, authored or reviewed drafts of the article, and approved the final draft.
- Xinghui Wu performed the experiments, analyzed the data, prepared figures and/or tables, and approved the final draft.
- BaoJun Yu performed the experiments, analyzed the data, prepared figures and/or tables, and approved the final draft.
- Ronglin Chen performed the experiments, analyzed the data, authored or reviewed drafts of the article, and approved the final draft.
- JieFu Lu analyzed the data, prepared figures and/or tables, and approved the final draft.
- Huasheng Tong conceived and designed the experiments, performed the experiments, analyzed the data, authored or reviewed drafts of the article, and approved the final draft.

### Animal Ethics

The following information was supplied relating to ethical approvals (i.e., approving body and any reference numbers):

The animal ethics committee of the General Hospital of Southern Theatre Command of PLA approved this study (SCXK, Guangdong 2016-0041).

### Data Availability

The raw measurements are available in the Supplemental Files.

### Supplemental Information

Supplemental information for this article can be found online at http://dx.doi.org/10.7717/peerj.13799#supplemental-information.

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
