# Peer review of "HMGB1-activatied NLRP3 inflammasome induces thrombocytopenia in heatstroke rat"

_PeerJ, doi:10.7717/peerj.13799_

## Round 0.1 · original submission · Major Revisions

As you will see, both reviewers felt this work has merit and have made positive comments. However, reviewer-2 in particular has raised a few points that need clarification and some suggestions for manuscript style should also be considered. Please address these points and resubmit, together with a detailed response to all points raised.

Reviewer 1 ·

Basic reporting

The abstract is poorly written in places - especially the methods - should replace "reflected" with a more appropriate term such as measured, analysed.

The introduction provides a good analysis of the background but doesn't really explain why it is important. The aims and hypothesis need to be more explicitly stated

Experimental design

The experiments are appropriate to address the research question and well designed and performed to a high technical ad ethical standard.

Some clarification is required regarding the specificity of the reagents used - e.g. are the neutralising/blocking antibodies specific for rats.

Validity of the findings

The data is clearly presented and described

Experiments were only conducted once but appear to be robust

The authors must justify the statistical analysis used. Can they confirm if analysis of the normality of the data performed. Given that small group sizes (n=6) were used it would be more appropriate to use non- parametric tests such as Mann Whitney and Kruskall- Wallis.

The majority of the discussion and conclusions are clear and well stated. However some paragraphs need some attention (starting line 346 and 354) as it is not clear what point is being made.

Additional comments

A well executed study with interesting results however it could explain what the implications of the work is and how it could be used to develop therapies.

The English requires some attention in the Abstract, Introduction and Discussion - it would improve the clarity of manuscript. Some odd phrases e.g. line 296.

Could include some comment of the limitations of the study

Reviewer 2 ·

Basic reporting

The manuscript conforms to the guidelines of the journal. Figure legends contain special characters (perhaps asterisks?) that have not come through in the PDF for review and are shown as boxes. This should be corrected. Additionally, figure legends should clearly state the sample size for each experiment, including both independent experiments and technical replicates in order to conform with usual standards and make the figures interpretable.

Experimental design

Methods and experimental design are relevant and well described.

Validity of the findings

The authors submit and interesting set of studies regarding the role of inflammasome activation in the development of thrombocytopenia in a rat model of heat stroke. This is a relevant problem which merits study. The authors show that induction of heat stroke (HS) in rats results in thrombocytopenia and platelet activation (CD62P expression). They demonstrate that in HS, platelets up regulate NLRP3 and Asc, and that cleaved caspase-1 levels are increased as well. Blockade of NLRP3 with MCC950, as expected, decreases cleaved caspase-1 levels, but also down regulates IL-1B expression. HMGB1 levels were elevated in HS, and blockade of HMGB1 with ethyl pyruvate (non-specific) or monoclonal antibody reduced NLRP3 expression and cleaved caspase-1 in platelets. This effect was found to be both via TLR4 and RAGE, and led to increased ROS production in platelets. While these findings are of interest, I have several reservations related to both mechanistic concerns and synthesizing these data into an overall model.

1. HS induces a decrease in platelet count, as shown. However, the authors discuss in discussion that HS can induce platelet clustering. This is not unexpected in a condition of significant inflammation and platelet activation. Mechanistically and clinically, an absolute loss of platelet number is not the same as platelet clumping. Did significant platelet aggregation occur here? This could be assessed by examining a slide from the peripheral blood, or even by simple spectrophotometry of the platelet-rich plasma to differentiate between cell loss and cell clumping.

2. The electronic microscopy images are very nice and illustrate interesting ultrastructural changes in thee platelets in HS. However, these data are really not integrated into the overall manuscript. The images are described, but the significance of these findings in the context of inflammasome activation or HS are not mentioned at all. If they cannot be linked to the system being studied, I would suggest they be removed.

3. My next concern is in regard to much of the inflammasome activation data and interpretation. In general, inflammasome activation requires 2 steps (see many reviews on the topic). Step 1, priming, is generally mediated by TLR4 and results in the upregulation of inflammasome machinery (NLRP3, Asc, pro-IL-1B and pro-caspase-1). Signal 2, which can take on many forms (membrane damage, potassium depletion, ionophores, and others) then triggers the inflammasome, resulting in Asc clustering, pro-caspase-1 cleavage, and subsequent cleavage of pro-IL-1B and gasdermin D both to active forms. These two events can be separated temporally. Further, in some conditions, the resulting trigger will result in cell lysis via pyroptosis. The data shown in this manuscript seem to relate mostly to priming. Upregulation of NLRP3 (fig 3A) and upregulation of pro-IL-1B (Fig 3B. 4E. and others; assuming the antibody used for flow is not specific to cleaved, mature IL-1B). Then, caspase-1 cleavage is indicative of inflammasome triggering.
It is unclear how the stimuli and inhibitors used in these studies can achieve both of these endpoints. For instance, MCC950 should reduce cleaved caspase-1 (as it does here), but why does caspase-1 inhibition with ac-YVAD-cmk down regulate IL-1B? Upregulation of pro-IL-1B is upstream of caspase-1 activation. This needs to be reconciled.

4. Regarding HMGB1 data - is a source of HMGB1 known or can it be speculated? Again, HMGB1, as a TLR4 agonist, is a strong primer of NLRP3 inflammasomes. But by itself is not a trigger of inflammasomes. Upregulation of NLRP3, as in point 3 above, needs to be separated from caspase-1 activation. IL-1B cleavage (a downstream event from NLRP3) or secretion (also downstream) are never assessed in the manuscript. I would suggest that assessment of plasma IL-1B levels, or at least immunoblotting to show cleavage of pro-IL-1B, would greatly enhance the ability of the authors to distinguish these two events.

5. Pyroptosis is mentioned in the discussion and is important here. Recent work (Volchuk et al Nature Communications) has shown that in the absence of pyroptosis, inflammasome activation itself does not lead to HMGB1 release. The authors state that since platelets do not have nuclei, that pyroptosis cannot be assessed, but this is not true. As they mention, a combination of caspase-1 cleavage and PI uptake will suffice. PI uptake occurs through gasdermin D in the plasma membrane, and/or through large PM holes induced by lytic death. This does not require a nucleus, and can be assessed in platelets.

---

## Round 0.2 · accepted · Accept

The reviewers are satisfied with your rebuttal and changes, so I am pleased to accept the paper now.

Reviewer 2 ·

Basic reporting

My issues have been addressed.

Experimental design

No issues.

Validity of the findings

My concerns on initial review have been addressed.